# A Novel Vectorized Curved Road Representation Based Aerial Guided Unmanned Vehicle Trajectory Planning

**DOI:** 10.3390/s23167305

**Published:** 2023-08-21

**Authors:** Sujie Zhang, Qianru Hou, Xiaoyang Zhang, Xu Wu, Hongpeng Wang

**Affiliations:** 1Tianjin College, University of Science & Technology Beijing, Tianjin 301830, China; m202220803@xs.ustb.edu.cn; 2Institute of Intelligence Technology and Robotic Systems, Shenzhen Research Institute of Nankai University, Shenzhen 518083, China; houqianru@nuaa.edu.cn (Q.H.); zxy@mail.nankai.edu.cn (X.Z.); 3College of Artificial Intelligence, Nankai University, Tianjin 300353, China; 4School of Traffic and Transportation, Beijing Jiaotong University, Beijing 100044, China; wuxu@bjtu.edu.cn; 5Laboratory of Science and Technology on Integrated Logistics Support, National University of Defense Technology, Changsha 410073, China

**Keywords:** vectorized curvedroad representation, GA-Bézie algorithm, route-plane, trajectory planning

## Abstract

Unmanned vehicles frequently encounter the challenge of navigating through complex mountainous terrains, which are characterized by numerous unknown continuous curves. Drones, with their wide field of view and ability to vertically displace, offer a potential solution to compensate for the limited field of view of ground vehicles. However, the conventional approach of path extraction solely provides pixel-level positional information. Consequently, when drones guide ground unmanned vehicles using visual cues, the road fitting accuracy is compromised, resulting in reduced speed. Addressing these limitations with existing methods has proven to be a formidable task. In this study, we propose an innovative approach for guiding the visual movement of unmanned ground vehicles using an air–ground collaborative vectorized curved road representation and trajectory planning method. Our method offers several advantages over traditional road fitting techniques. Firstly, it incorporates a road star points ordering method based on the K-Means clustering algorithm, which simplifies the complex process of road fitting. Additionally, we introduce a road vectorization model based on the piecewise GA-Bézier algorithm, enabling the identification of the optimal frame from the initial frame to the current frame in the video stream. This significantly improves the road fitting effect (EV) and reduces the model running time (*T*—model). Furthermore, we employ smooth trajectory planning along the “route-plane” to maximize speed at turning points, thereby minimizing travel time (*T*—travel). To validate the efficiency and accuracy of our proposed method, we conducted extensive simulation experiments and performed actual comparison experiments. The results demonstrate the superior performance of our approach in terms of both efficiency and accuracy.

## 1. Introduction

Real-time vision sharing between drones and unmanned ground vehicles is a critical technology for enabling collaborative tasks in heterogeneous multi-domain unmanned systems, particularly in scenarios where GPS and other commercial communication networks are unavailable. Examples of such scenarios include passable area detection for polar scientific research and field ecological monitoring. In recent years, advancements in intelligent driving, road extraction, and trajectory planning technologies have expanded the scope of research in trajectory planning for ground-based mobile robots. Unmanned ground vehicles frequently encounter the challenge of traversing mountainous terrains characterized by numerous unknown continuous curves. The intricate nature of these paths, combined with overgrown vegetation that obscures road boundaries and obstructed field of vision due to trees, results in an unstructured path.

These problems can be effectively addressed through the collaborative efforts of multiple robots operating in open spaces. Under the guidance of aerial vision, unmanned vehicles can accomplish tasks with enhanced efficiency. The wide viewing angles and vertical displacement capabilities of aerial robots [1,2] offer a viable solution to compensate for the limited field of view inherent in ground vehicles [3].

Gaining a comprehensive understanding of the feasible travel area is an essential prerequisite in various applications [4]. The establishment of an environment model is a fundamental requirement for optimizing time and energy consumption in trajectory planning. Ground vehicles face limitations in their field of view, which affects their ability to plan trajectories. To ensure safety in visual blind spots, they need to reduce their velocity. In contrast, aerial vehicles can provide a broader field of view, enabling ground vehicles to anticipate road conditions in advance. Owing to the inherently rigid nature of smart vehicles, their motion needs to be constrained to ensure a continuous and smooth trajectory. Once the road area is detected, it is essential to extract and fit the road lines into an appropriate mathematical model. This model, when combined with motion constraints, can determine a suitable reference trajectory for the vehicle.

Ground vehicles inherently suffer from a restricted field of view, which poses challenges in their ability to perceive and navigate their surroundings effectively. Moreover, conventional road line extraction methods solely provide pixel-level positional information, limiting their utility in trajectory planning. Simultaneously, when unmanned aerial vehicles guide unmanned ground vehicles through the transmission of image sequences, the processing time and accuracy of image analysis and road fitting are often suboptimal. To address these limitations, we propose a visually guided trajectory planning method for unmanned ground vehicles that incorporates a road route fitting approach. As shown in Figure 1, the primary objective of this study is to utilize drones or other aerial vehicles to acquire information about the drivable area for ground vehicles. Subsequently, an algorithm is employed to identify road surfaces and extract road lines, which are represented as segmented Bézier curves. By incorporating road parameters, including length, curvature, and corners, along with the principles of instantaneous center theory in rigid body motion, the trajectory planning for the ground vehicle is determined. The key contributions of this research can be summarized as follows:

1.We propose a novel sorting model based on the K-Means clustering algorithm, as well as a road vectorization model based on the piecewise GA-Bézier algorithm. These models offer significant improvements in fit quality and run time, enabling the realization of continuous tangents and continuous curvatures in the road representation.2.We propose an advanced trajectory planning algorithm that operates along the “route-plane” to optimize the travel time of the ground vehicle. This algorithm takes into account various constraints, including velocity and acceleration limitations, to ensure the feasibility and safety of the planned trajectory. By optimizing the travel time while adhering to these constraints, our algorithm aims to enhance the overall efficiency and effectiveness of the ground vehicle’s movement.3.We propose three evaluation indices to assess the performance of our method: the road fitting effect (*EV*), the model running time (*T—model*), and the travel time (*T—travel*). These indices serve as quantitative measures to evaluate the effectiveness, efficiency, and accuracy of our proposed method. To validate the performance of our approach, we conducted extensive simulations and practical experiments. The results of these experiments demonstrate the efficiency and accuracy of our proposed method, further affirming its effectiveness in real-world scenarios.

This paper serves as an extension of our previously published conference paper [5], aiming to further enhance the optimization of planned trajectories for ground vehicles. The rest of this paper is organized as follows. Section 2 provides an overview of the related work in the field. Section 3 formulates the problem and evaluation metrics. Section 4 introduces the road extraction and star points sorting method employed to accurately identify and represent road surfaces. Section 5 presents the traditional road geometric parameter model. Section 6 presents the piecewise GA-Bézier road vectorization model, which enables the efficient representation of road lines. Section 7 details the trajectory planning method, incorporating road parameters and the principles of instantaneous center theory to optimize the ground vehicle’s trajectory. Section 8 describes the experimental verification conducted to evaluate the proposed approach. Finally, Section 9 concludes the paper, summarizing the key findings and discussing potential avenues for future research.

## 2. Related Work

Numerous research studies have employed onboard cameras to effectively identify the unobstructed road ahead of a vehicle, a crucial requirement for intelligent assisted driving systems. While the approach of extracting road information using a local field of view significantly differs from that of utilizing a global field of view, there are certain shared characteristics between the two methodologies. Road extraction techniques often rely on color and texture analysis [6,7]. However, in complex and unstructured environments, the wide range of texture and color characteristics poses challenges in accurately differentiating roads from their surroundings. The variability in road colors further complicates the task, making it impractical to assign a fixed color for road representation. Alternatively, detecting road boundaries and fitting them to a model is another approach [8]. Nonetheless, this method proves inadequate when dealing with ambiguous boundaries, as is often the case with unstructured roads. A third avenue for road detection involves leveraging vanishing points [9,10]. However, this approach proves ineffective in scenarios where a clear vanishing point is absent or when dealing with curved road boundaries. To address the challenges posed by abrupt variations in road conditions, certain researchers have advocated for the integration of prior information with road detection techniques. For instance, Alvarez et al. proposed a methodology that combines prior information derived from a geographic information system (GIS) with the estimated road derived from the current image, thereby achieving robust road segmentation [11,12]. Nevertheless, it is worth noting that the efficacy of their approach is contingent upon the availability of a GIS database. In the absence of such a database, their method may encounter limitations.

The aforementioned road detection methods primarily revolve around pixel classification to distinguish between road and non-road regions. Once the road area is successfully identified, the subsequent step involves extracting the road lines to establish a reference for the roadbed. Road line detection and centerline extraction are the two prevailing techniques employed for directly obtaining the centerline of a road from a global image [10,13,14,15].

Song and Civco [16] introduced a two-step approach for road centerline extraction. Their method employs a support vector machine (SVM) to differentiate the road from the background, followed by utilizing a shape index to detect road lines. Gamba et al. [17] proposed a comprehensive algorithm specifically designed for urban areas, which incorporates adaptive filtering to determine the primary direction of a road. Subsequently, a perceptual grouping algorithm is employed to eliminate redundant road segments and bridge any existing gaps. By examining road intersections and standardizing the overall pattern, the road network topology can be identified. Huang et al. [18] presented a road centerline extraction technique that combines multiscale information with SVM. The structural characteristics of the mixed spectrum are analyzed using an SVM classifier, and the classification results from multiple scales are merged. Finally, a morphological refinement algorithm is applied to accurately detect the road centerline.

The road detection data obtained through multi-agent cooperation are derived from aerial and satellite imagery [19]. Tan et al. [20] introduced an iterative graph exploration scheme for automatic road network map extraction from aerial maps. This approach incorporates adaptive step size and leverages road segmentation results as prior information to guide the expansion of road network maps. Addressing the limitations of path extraction based on deep learning, such as the reliance on multiple convolution operators, challenges in accurately predicting contextual spatial relationships, inadequately labeled data, and limited model portability, Zhu et al. [21] proposed a novel road extraction framework known as GCB-Net (Global Context-aware and Batch-independent Network). GCB-Net enhances the encoder–decoder structure by incorporating a global context awareness (GCA) block, which effectively integrates global context features. To enhance the original basic network, the Filter Response Normalization (FRN) layer is employed, eliminating batch dependence, accelerating the learning process, and further enhancing the model’s robustness.

While the aforementioned methods excel in accurately extracting road lines based on pixel positions, they fall short in providing sufficient information to support ground vehicle trajectory planning. Representing the road line solely by the width of a single pixel lacks the necessary details for formulating a comprehensive mathematical model that adheres to constraints such as continuous curvature for the lines. Therefore, additional processing steps are required to enhance the extracted road lines, enabling the formulation of a mathematical model that satisfies the aforementioned constraints.

## 3. Problem Formulation and Evaluation Index

### 3.1. Assumptions

Consider a scenario where a gimbal camera mounted on a quadrotor drone is used to photograph an image containing road and non-road areas. We assume the following:1.The pan-tilt (PT) camera has been pre-calibrated, and radial distortion is eliminated.2.The orientation of the PT camera is controlled by the gimbal stabilizer, so ground images can be captured steadily and reliably.3.The road has several feature points that can be captured by the camera and used to convert the camera coordinate system to the global coordinate system.

### 3.2. Problem Statement

The objective of this study was to use aerial vision to take pictures of unknown roads, use image processing algorithms to extract and detect road lines, and use a predesigned mathematical model called a piecewise GA-Bézier curve for road fitting. After the road model parameters are optimized, they are mapped to the global coordinate system, and the trajectory is planned according to the kinematic model and constraints of the ground vehicle so that it can pass over the road quickly, efficiently, and safely.

**Definition 1.** 
*The piecewise GA-Bézier curve is a segmented Bézier curve using a genetic algorithm that stitches piecewise according to the constraints on the ending point. Each ending point should meet the constraints of a continuous position, continuous tangent, and continuous curvature.*


Given a set of aerial images Img, an image detection algorithm can be used to extract the road surface as the discrete point set *S*. The objective of trajectory fitting and planning is given by
(1)$$\begin{array}{c} \min\;\;\;J(S)\\ \min\;\;\;T(P(t))\\ \;s.t.\;\;\;\;g(S,P(t)) \end{array}$$
where P(t) is the road model (i.e., piecewise GA-Bézier curve) fitted to the scattered point set *S*. The task is to minimize the fitting error *J* and travel time *T* of the ground vehicle while satisfying the relevant constraints g(S,P(t)). The detailed formulation of the objectives and constraints is given in the following sections.

**Definition 2.** 
*By establishing reference points and camera parameters within the world coordinate system, the image can be effectively rectified through the application of perspective and affine transformations.*


This process involves directly transforming the coordinate system based on the correspondence between reference points in the image and their corresponding positions in the world coordinate system. Such an approach ensures accurate correction of the image, enhancing its visual representation and aligning it with the desired perspective. During the practical experimentation phase, the measurements are conducted within the designated range of the world coordinate system. To facilitate accurate correction, reference points are strategically positioned at the initiation and termination points of the road, which are situated at diagonally opposite corners of the aforementioned rectangular area. By applying rotation and translation transformations, the original image data can be effectively rectified and aligned with the world coordinate system. This process ensures the preservation of spatial accuracy and facilitates meaningful analysis and interpretation of the corrected data.

### 3.3. Evaluation Index

Three evaluation indices were devised to assess the performance of the proposed road fitting method. The first index, denoted as EV, was employed to evaluate the road error. The second index, referred to as *T*—model, was utilized to evaluate the efficiency achieved by the road fitting method. Additionally, the driving time of the unmanned vehicle, obtained through the proposed trajectory planning method, was evaluated using the third index, denoted as *T*—travel.
(2)$$\begin{array}{c} \begin{array}{c} EV=\max_{i}\left(\min_{m}\left(∥S_{i}-P\left(m\right)∥\right)\right) \end{array} \end{array}$$

In Equation (Equation 2), let *i* represent the count of sorted road stars, *m* denote the number of Bessel curve points, and $S_{i}$ represent the star point data in paragraph *i*, which are obtained through stepwise sorting after the initial road stars are acquired via image processing. Furthermore, P(m) represents the coordinates of the point on the fitted Bézier curve.

The term *T*—model refers to the time required by both the proposed method and the traditional method for the task of road star extraction and fitting, assuming an identical computer configuration.
(3)$$\begin{array}{c} \begin{array}{c} t\left(s\right)=\int_{0}^{s}\frac{1}{v\left(S\right)}dS \end{array} \end{array}$$

The variable *T*—travel represents the travel time of the driverless car, which is determined by integrating the velocity associated with each point along the Bézier curve.

## 4. Road Extraction and Star Points Sorting

In outdoor environments, the road surface typically exhibits a relatively flat profile compared to its surroundings. When captured from an aerial perspective, the road surface tends to display a lower degree of diffuse reflection, a higher degree of specular reflection, increased brightness, and enhanced flatness in comparison to the surrounding areas. These characteristics contribute to a smoother texture appearance. Figure 2 illustrates our devised approach for extracting discrete points, referred to as “star points”, from a captured image, specifically targeting the road surface.

As shown in Figure 3a, the star points, which are directly extracted from the image, follow a trajectory that aligns with the lines present in the photograph. To potentially optimize the fitting accuracy and improve computational efficiency, we have devised a stepwise sorting method based on the K-Means clustering algorithm. The fundamental concept behind the K-Means algorithm is to iteratively determine a partition scheme consisting of K clusters, aiming to minimize the corresponding loss function. This loss function is defined as the sum of squared errors, which quantifies the distance between each sample and the centroid of the cluster to which it belongs.
(4)$$J\left(c,\mu \right)=\sum_{i=1}^{M}{∥x_{i}-\mu_{c_{i}}∥}^{2}$$
where $x_{i}$ stands for sample *i*, $c_{i}$ is the cluster to which $x_{i}$ belongs, $\mu_{c_{i}}$ represents the center of the cluster, and *M* is the total number of samples.

By incorporating the K-Means clustering algorithm prior to the sorting algorithm, the number of samples is effectively reduced, resulting in improved operational efficiency. This is in accordance with Equation (Equation 4). The stepwise sorting method, which utilizes the K-Means clustering algorithm, facilitates the systematic arrangement of star points along the road line. The algorithm initiates from the first star point and identifies the closest remaining star point as the target. This target star point is then appended to the sorted list, becoming the new current star point, and the search continues through the unsorted list. As there may be multiple target star points equidistant to the current star point in the unsorted list, any missed star points are subsequently revisited during the sorting process. To ensure efficient sorting, a threshold value is established. If the distance between the target and current star points exceeds this threshold, the sorting process terminates promptly. Although the sorting algorithm may overlook certain star points, the overall impact is negligible. Figure 3c visually depicts the sorted star points, which accurately align with the road line.

To establish the universality of the road extraction and star point sorting methods, we conducted a series of experiments encompassing various scenarios. The final column in Figure 4 showcases the sorted star points, which exhibit a remarkable alignment with the road.

## 5. Traditional Road Geometric Parameter Model

The traditional road geometric parameter model divides the structured road into *i* road units, and each unit is composed of all or part of the elements in the straight line $L_{i}$, the cyclograph $C_{i1}$, the cyclograph $C_{i2}$, and the arc $S_{i}$, namely $“L_{i}+C_{i1}+S_{i}+C_{i2}”$. A continuous free curve is created by seamlessly connecting multiple route units. Subsequently, the transverse direction of the free curve is fixed to generate a fixed-width road model. The data storage format for each unit can be represented by the following Equation (Equation 5):(5)$$U_{i}=\left\{i,l_{i},\theta_{C_{i1}},R_{i},\theta_{s_{i}},\theta_{C_{i2}},\pm 1\right\}\;$$
where *i* represents the route unit, which is the *i* unit of the entire route, $l_{i}$ represents the length of the straight line element $L_{i}$, $R_{i}$ represents the radius of the arc element $S_{i}$, ±1 represents the entire unit’s deflection direction, where +1 represents the clockwise deflection and −1 represents the counterclockwise deflection, $\theta_{C_{i1}}$ represents the deflection angle of cyclotron $C_{i1}$, $\theta_{s_{i}}$ represents the deflection angle of arc element $S_{i}$, and $\theta_{C_{i2}}$ represents the deflection angle of cyclotron $C_{i2}$. Figure 5 shows a route unit model.

Once the road skeleton diagram is obtained, the line positions can be determined using the Hough transformation principle. By analyzing the line elements, it becomes evident that there are two cyclotron lines and an arc between them. Additionally, the outermost boundary may consist of one or more of the aforementioned elements. To establish the boundary, a cubic function curve is fitted to each segment of discrete data, with a straight line connecting the infinite curvature radius at the center of symmetry and one end of the circular arc. The fitting process ensures that the coefficient of the cubic function maintains the curvature continuity of the entire curve, specifically at the point of contact with the circular arc, where the curvature radius matches the radius of the arc precisely.

Drawing upon Sichuan road images as a case study, road fitting is conducted utilizing the conventional road geometric parameter model. The fitting results are visually depicted in Figure 6, wherein the green curve represents a cyclotron, the black curve represents an arc, and the red line segment represents a straight line. The corresponding geometric parameters extracted from this image are presented in Table 1, providing a comprehensive overview of the road’s spatial characteristics and enabling further quantitative analysis and evaluation.

## 6. Piecewise GA-Bézier Road Vectorization Model

Each road unit is modeled as a fifth-order Bézier curve P(m) for which $P_{0},P_{1},P_{2},P_{3},$$P_{4}$, and $P_{5}$ are the control points represented by coordinating column vectors. Let *m* be an increasing variable in the interval [0,1]. The matrix equation for calculating the trajectory according to the control points and *m* is given by Equation (Equation 6), where the control points $Pcs=\left[\begin{array}{cccccc} P_{0} & P_{1} & P_{2} & P_{3} & P_{4} & P_{5} \end{array}\right]$ and the constant $C=diag(\left[\begin{array}{cccccc} 1 & 5 & 10 & 10 & 5 & 1 \end{array}\right])$. R(m) is a six-dimensional column vector of the combined power of *m*, which means that $R{\left(m\right)}_{i}=m^{i}{\left(1-m\right)}^{5-i},\;\left(i=0,1,2,3,4,5\right)$.
(6)$$\begin{array}{c} P(m)=Pcs*C*R(m) \end{array}$$

When the control points align in a collinear manner, the fifth-order Bézier curve undergoes degeneration, resulting in segmented curves. In this degenerate state, the radius of curvature becomes infinite, and the curvature itself becomes zero. This characteristic aligns with the behavior of straight line segments, where the curvature is inherently absent.
(7)$$\begin{array}{c} P^{\prime }\left(m\right)=Pcs*C*R^{\prime }\left(m\right)\\ P^{\prime \prime }\left(m\right)=Pcs*C*R^{\prime \prime }\left(m\right)\\ K\left(m\right)=\frac{\left|\right|P^{\prime }\left(m\right)\times P^{\prime \prime }\left(m\right)\left|\right|}{\left|\right|P^{\prime }\left(m\right)|{|}^{3}} \end{array}$$

In order to satisfy the condition of continuous curvature, it is necessary for the bending curvature of the preceding road unit to be equal to the initial curvature of the subsequent road unit at their junction point. This ensures a smooth transition between the two road units, maintaining a seamless curvature profile throughout the road network. The curvature of the fifth-order Bézier curve K(m) is calculated using Equation (Equation 7).

The road line is expanded along the positive and negative normal directions of each point to form a road. If the free variable $P_{f}\left(m\right)$ is the tangent direction, it can be turned counterclockwise 90° to obtain the negative normal direction $N_{n}\left(m\right)$ and turned clockwise 90° to obtain the positive normal direction $N_{p}\left(m\right)$. The rotation matrix method T(θ) in Equation (Equation 8) is used for the calculation.
(8)$$\begin{array}{c} P_{f}\left(m\right)=\frac{P^{\prime }\left(m\right)}{\left|\right|P^{\prime }\left(m\right)\left|\right|}\;T\left(\theta \right)=\left[\begin{array}{cc} \cos(\theta ) & -\sin(\theta )\\ \sin(\theta ) & \cos(\theta ) \end{array}\right]\\ N_{n}\left(m\right)=T\left(\frac{\pi }{2}\right)*P_{f}\left(m\right)\;N_{p}\left(m\right)=T\left(-\frac{\pi }{2}\right)*P_{f}\left(m\right) \end{array}$$

If the width of the road is W(m), the coordinates of the positive side of the road are $P_{p}\left(m\right)$, and the coordinates of the negative side of the road are $P_{n}\left(m\right)$, the calculation formula is as follows:(9)$$\begin{array}{c} P_{n}\left(m\right)=P\left(m\right)+0.5\cdot W\left(m\right)\cdot N_{n}\left(m\right)\\ P_{p}\left(m\right)=P\left(m\right)+0.5\cdot W\left(m\right)\cdot N_{p}\left(m\right) \end{array}$$

Suppose that the starting point of a certain road unit is $P_{s}$, the ending point is $P_{e}$, the initial curvature is $K_{s}$, and the ending curvature is $K_{e}$. Then,
(10)$$\begin{array}{c} P\left(0\right)=Ps=P_{0}\;\;\;\;P\left(1\right)=Pe=P_{5}\\ Ks=\frac{\left|\right|P^{\prime }\left(0\right)\times P^{\prime \prime }\left(0\right)\left|\right|}{\left|\right|P^{\prime }\left(0\right)|{|}^{3}}\;Ke=\frac{\left|\right|P^{\prime }\left(1\right)\times P^{\prime \prime }\left(1\right)\left|\right|}{\left|\right|P^{\prime }\left(1\right)|{|}^{3}} \end{array}$$

When $K_{s}$ is infinite, then $P_{0}$ and $P_{1}$ are equal, and $P_{2}$ becomes a free variable. Similarly, when $K_{e}$ is infinite, then $P_{4}$ and $P_{5}$ are equal, and $P_{3}$ becomes a free variable.

As shown in Figure 7, given a set of specific parameters, several road units can be grown to form a complete road.

Due to the inherent complexity of road structures, achieving satisfactory results through the fitting of Bézier curves can be challenging. To address this issue, a sequential division approach is employed, wherein the *n* star points are divided into *N* segments. Each segment is then individually fitted to a fifth-order Bézier curve. This method allows for a more accurate representation of the intricate road geometry, enhancing the overall quality of the curve fitting process. The threshold Re is set to determine the beginning and end of each segment. The initial demarcation point is taken as the $\left[\begin{array}{cccccc} 1 & n/N & 2n/N & 3n/N & \cdot \cdot \cdot & n \end{array}\right]$ star point. Then, a circle is drawn with the initial demarcation point as the center and Re as the radius. Then, the average of the star point coordinates within the circle is taken to determine the standard demarcation point. The starting and ending points of each segment of the road line are given by $\left[\begin{array}{cccccc} {}^{0}X & {}^{1}X & {}^{2}X & {}^{3}X & \cdot \cdot \cdot & {}^{N}X \end{array}\right]$. The initial demarcation point does not necessarily correspond to the average of the number of star points. Instead, the demarcation parameters can be preconfigured to obtain non-uniform points. To establish the standard demarcation point, a circle with a radius of Re is employed, derived from the initial demarcation point.

Consider a specific segment of a segmented road for illustration purposes. The process of Bézier curve fitting can be conceptualized as an optimization problem, which can be mathematically represented by the following Equation (Equation 11). In practice, if the *k*th road has $n_{k}$ star points and the corresponding control variables for the Bézier curve are divided into $l_{k}$ parts, the distance cost matrix *D* can be constructed as given in Equation (Equation 12).
(11)$$\begin{array}{c} \min_{P(\cdot )}J=\max_{i}(\min_{m}\left(|\right|S_{i}-P\left(m\right)||))\\ \;\;\;+\max_{m}(\min_{i}\left(|\right|S_{i}-P\left(m\right)||)) \end{array}$$
where *i* represents sorted road star points, *m* represents Bessel curve points, $S_{i}$ represents the star point data in paragraph *i* obtained by stepwise sorting after obtaining the original road star points through image processing, and P(m) are the coordinates of the points on the fitted Bézier curve.
(12)$$D=\left[\begin{array}{ccc} \left|\right|S_{1}-P\left(m_{1}\right)\left|\right| & \cdot \cdot \cdot & \left|\right|S_{n_{k}}-P\left(m_{1}\right)\left|\right|\\ \left|\right|S_{1}-P\left(m_{2}\right)\left|\right| & \cdot \cdot \cdot & \left|\right|S_{n_{k}}-P\left(m_{2}\right)\left|\right|\\ \cdot \cdot \cdot & \cdot \cdot \cdot & \cdot \cdot \cdot \\ \left|\right|S_{1}-P\left(m_{l_{k}}\right)\left|\right| & \cdot \cdot \cdot & \left|\right|S_{n_{k}}-P\left(m_{l_{k}}\right)\left|\right| \end{array}\right]$$

In the distance cost matrix *D*, the minimum value of each column is taken to obtain a row vector. Then the largest element in the row vector is designated as $M_{s}$. Then the minimum value from each row of the matrix *D* is taken to obtain a column vector, and the largest element in the column vector is designated as $M_{p}$. The sum of $M_{s}$ and $M_{p}$ is the performance metric *J*. The purpose of this setup is to minimize the distance between the star point and curve and to minimize the occurrence of local self-intersections for the curve.

The parameter matrix of road units is given by ${}^{k}Pcs=\left[\begin{array}{cccccc} {}^{k}P_{0} & {}^{k}P_{1} & {}^{k}P_{2} & {}^{k}P_{3} & {}^{k}P_{4} & {}^{k}P_{5} \end{array}\right]$. It is necessary to ensure a continuous tangent and continuous curvature at each junction.

For the first road unit k=1, the starting point is ${}^{0}X$, and the ending point is ${}^{1}X$. Let ${}^{1}P_{0}={}^{0}X\;and\;{}^{1}P_{5}={}^{1}X$, then ${}^{1}P_{f}=\left[\begin{array}{cccc} {}^{1}P_{1} & {}^{1}P_{2} & {}^{1}P_{3} & {}^{1}P_{4} \end{array}\right]$ becomes the free variable, and it can be optimized by a genetic algorithm according to the performance index ${}^{1}J$. Due to the inherent limitations of genetic algorithms in locating the global optimal solution and the potential occurrence of curve self-intersections during the fitting process, it becomes imperative to appropriately configure the genetic variables of individuals within the population to mitigate such occurrences. This measure aims to minimize the likelihood of encountering self-crossing curves and enhance the overall effectiveness of the genetic algorithm in achieving optimal solutions.

As shown in Figure 8, the four intermediate control points are characterized by chord lengths and angles in relation to the initial and terminal points. This representation method enables a more precise and structured depiction of the control points, facilitating a comprehensive understanding of their spatial arrangement and geometric properties.

The genetic variables $M_{L\theta }$ are set to the elements in the matrix as follows:(13)$$M_{L\theta }=\left[\begin{array}{cccc} L_{01} & L_{12} & L_{34} & L_{45}\\ \theta_{01} & \theta_{12} & \theta_{34} & \theta_{45} \end{array}\right]$$

The parameters $L_{D}$ and $\theta_{D}$ are used to set the initial search range of genetic variables so that a control point moves as far as possible along with the star point:(14)$$\begin{array}{c} L_{D}=\left|\right|P_{5}-P_{0}\left|\right|\\ \theta_{D}=\mathrm{atan}2\left(P_{5y}-P_{0y},P_{5x}-P_{0x}\right) \end{array}$$

In the utilized genetic algorithm, consecutive decision variables are encoded using arithmetic scales and Gray code, guaranteeing precision up to six decimal places. For binary chromosomes, a two-point crossover is employed, with the selection process being based on random sampling. The mutation probability is determined as the reciprocal of the chromosome length, while the crossover probability is set to 0.9. This particular genetic algorithm incorporates elitist reservations, thus earning it the designation of Elitist Reservation GA. By incorporating these techniques and strategies, the algorithm aims to enhance the accuracy, efficiency, and effectiveness of the optimization process.

After the optimal solution is obtained, the tangent direction ${}^{1}V$ and curvature ${}^{1}K$ at the ending point are calculated according to Equation (Equation 7).
(15)$$\begin{array}{c} {}^{k-1}V//\left({}^{k}P_{1}-{}^{k}P_{0}\right)\\ {}^{k-1}K=\frac{\left|\right|{}^{k}P^{\prime }\left(0\right)\times {}^{k}P^{\prime \prime }\left(0\right)\left|\right|}{\left|\right|{}^{k}P^{\prime }\left(0\right)|{|}^{3}} \end{array}$$

For the following road units, k=2,3,···,N, the starting point is ${}^{k-1}X$, and the ending point is ${}^{k}X$. Let ${}^{k}P_{0}={}^{k-1}X\;and\;{}^{k}P_{5}={}^{k}X$. When the requirements for the junction between the starting point of the current road unit and ending point of the previous road unit are combined with Equation (Equation 7), the constraint conditions can be formulated as Equation (Equation 15).

Simplification shows that the constraints in Equation (Equation 15) involve $\left[\begin{array}{cc} {}^{k}P_{1} & {}^{k}P_{2} \end{array}\right]$ but not $\left[\begin{array}{cc} {}^{k}P_{3} & {}^{k}P_{4} \end{array}\right]$. Given two scalars of ${}^{k}L_{01}$ and ${}^{k}P_{2x}$, $\left[\begin{array}{cc} {}^{k}P_{1} & {}^{k}P_{2} \end{array}\right]$ can be solved using a linear equation. Therefore, when the auxiliary variable ${}^{k}P_{12}={\left[\begin{array}{cc} {}^{k}L_{01} & {}^{k}P_{2x} \end{array}\right]}^{T}$ is constructed, $\left[\begin{array}{ccc} {}^{k}P_{12} & {}^{1}P_{3} & {}^{1}P_{4} \end{array}\right]$ becomes a free variable, and it can be optimized using a genetic algorithm based on the performance index ${}^{k}J$.

Due to the constraints imposed by the initial curvature and initial tangent, expressing the chord length and angle of the first road unit would result in a significant number of NaN (not a number) values during iterative calculations. To overcome this limitation, we introduced auxiliary variables based on Cartesian coordinates. The Elitist Reservation GA is still employed, with the encoding method, variation method, variation probability, and crossover probability parameters set in a similar manner to those used for the first road unit.

After the optimal solution is obtained, the tangent direction ${}^{k}V$ and curvature ${}^{k}K$ at the ending point are calculated according to Equation (Equation 7).

The overall fitting is shown in Algorithm 1. Heuristic algorithms such as the genetic algorithm are often unable to obtain the global optimal solution directly. Thus, each frame of the aerial video stream is actually processed by the algorithm several times to select multiple local optimal solutions and approximate the global optimal solution.

After the algorithm is used to fit the road line segments to a Bézier curve, the road width *W* is calculated using Equation (Equation 16), where λ is the road coefficient.
(16)$$W=\lambda \cdot \max_{i}(\min_{m}\left(|\right|S_{i}-P\left(m\right)||))$$
Equation (Equation 9) can be used to extend the road lines on both sides. Figure 9 shows the road fitting results. The road extraction and fitting results matched the actual road and can be projected to the global coordinate system as the road model.
**Algorithm 1:** Fitting Road Line Segments**Input**: *N*, $sortStars=\left[\begin{array}{cccc} S_{1} & S_{2} & \cdot \cdot \cdot & S_{n} \end{array}\right]$**Output**: $\left[\begin{array}{ccccc} {}^{1}Pcs & {}^{2}Pcs & {}^{3}Pcs & \cdot \cdot \cdot & {}^{N}Pcs \end{array}\right]$$\left[\begin{array}{c} \left[\begin{array}{cc} {}^{0}X^{T} & n \end{array}\right]\\ \left[\begin{array}{cc} {}^{1}X^{T} & n_{1} \end{array}\right]\\ \cdot \cdot \cdot \\ \left[\begin{array}{cc} {}^{N}X^{T} & n_{N} \end{array}\right] \end{array}\right]$← divParts(sortStars,N);${}^{1}P_{0}$←${}^{0}X$;${}^{1}P_{5}$←${}^{1}X$;$\left[\begin{array}{cccc} {}^{1}P_{1} & {}^{1}P_{2} & {}^{1}P_{3} & {}^{1}P_{4} \end{array}\right]$← BézierFitGA($S_{1:n_{1}}$);${}^{1}Pcs$←$\left[\begin{array}{cccccc} {}^{1}P_{0} & {}^{1}P_{1} & {}^{1}P_{2} & {}^{1}P_{3} & {}^{1}P_{4} & {}^{1}P_{5} \end{array}\right]$;${}^{1}V$←${}^{1}P_{1}{-}^{1}P_{0}$;${}^{1}K$←$\frac{\left|\right|{}^{1}P^{\prime }\left(0\right)\times {}^{1}P^{\prime \prime }\left(0\right)\left|\right|}{\left|\right|{}^{1}P^{\prime }\left(0\right)|{|}^{3}}$;**for***each k in $\left[\begin{array}{cccc} 2 & 3 & \cdot \cdot \cdot & N \end{array}\right]$***do**
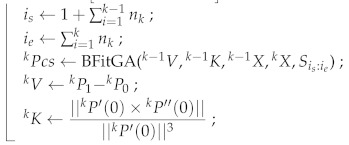
return $\left[\begin{array}{ccccc} {}^{1}Pcs & {}^{2}Pcs & {}^{3}Pcs & \cdot \cdot \cdot & {}^{N}Pcs \end{array}\right]$

## 7. Trajectory Planning

In order to ensure the stable operation of the robot, it is imperative to maintain a constant speed during the robot’s traversal along a curved arc. Specifically, the robot should neither experience acceleration nor deceleration while passing through the arc. However, the robot can be subjected to acceleration or deceleration when navigating along a straight rotation line. This approach ensures optimal performance and maneuverability of the robot during its operation. Therefore, in the context of trajectory planning for ground vehicles, it is crucial to consider the performance and road limitations. Specifically, it is imperative to ensure that the velocity of the vehicle does not surpass the predefined maximum velocity, denoted as $v_{M}$, and the maximum turning velocity, denoted as $v_{m}$, when the turning radius is *r*. By adhering to these constraints, we can guarantee the safe and efficient operation of the vehicle while navigating curves.

For trajectory planning, several fundamental requirements must be met. Firstly, the initial velocity at the starting point should be zero, and the velocity should decrease to zero at the ending point. Additionally, the velocity should exhibit continuous changes throughout the entire route, while ensuring that both the velocity and acceleration remain within predefined safety limits. Moreover, the maximum speed of the vehicle should not surpass the predefined threshold denoted as $v_{M}$. When the vehicle encounters a turn, it is crucial to adhere to a maximum speed limit denoted as $v_{m}$. The primary objective of trajectory planning is to obtain a trajectory that minimizes the overall travel time. Given that the velocity vector is always aligned with the tangent of the road, it suffices to plan the linear velocity at each point along the trajectory. By focusing on the linear velocity, we can effectively address the key aspects of trajectory planning while simplifying the computational complexity associated with considering the full velocity vector. This approach strikes a balance between accuracy and efficiency in trajectory planning.
(17)$$P\left(u\right)=\left\{\begin{array}{c} Pcs_{1}*C*R\left(u-1\right)\;\;u\in \left[1,2\right]\\ Pcs_{2}*C*R\left(u-2\right)\;\;u\in \left[2,3\right]\\ \ldots \ldots \\ Pcs_{N}*C*R\left(u-N\right)\;u\in \left[N,N+1\right] \end{array}\right.$$

Equation (Equation 17) shows the segmented road line model based on a fitted Bézier curve consisting of *N* segments where *u* is the extended control variable; it represents the control variable *m* for each segment of the curve, and is mapped to each point on the road line sequentially. Because each section of the road line meets the requirements of a continuous tangent and continuous curvature, the turning radius *r* at each point and the cumulative path length *S* can be calculated as follows:(18)$$r\left(u\right)=\frac{\left|\right|P^{\prime }\left(u\right)|{|}^{3}}{\left|\right|P^{\prime }\left(u\right)\times P^{\prime \prime }\left(u\right)\left|\right|}$$
(19)$$S\left(u\right)=\int_{1}^{u}\left|\right|P^{\prime }\left(z\right)\left|\right|dz$$

We can calculate the velocity limit $v_{p}$ at each point according to the following rules: |R| stands for radius of curvature. When 0<|R|<100 m, the value of $v_{p}$ remains unchanged. If 100 m <|R|<200 m, the upper limit is $v_{m}$. If the value exceeds 200 m, the value is $v_{m}$. When |R|>200 m, the further away the fitted curve is from the bending point, the speed of the unmanned vehicle takes $v_{M}$ as its upper limit, and anything exceeding $v_{M}$ is set to $v_{M}$. After the $v_{p}$ is integrated, the running time of the vehicle can be obtained. We can take the cumulative path length *S* as the abscissa and the squared velocity $v^{2}$ of each point as the ordinate to construct the route-plane. For any two points $\left(S_{A},v_{p_{A}}^{2}\right)$ and $\left(S_{B},v_{p_{B}}^{2}\right)$ on the $v_{p}^{2}-S$ curve, if the acceleration is uniform, then the movement from *A* to *B* is given by $v_{p_{B}}^{2}-v_{p_{A}}^{2}=2a\left(S_{B}-S_{A}\right)$.

When $\lim S_{A}-S_{C}\to 0_{-}\;and\;\lim S_{B}-S_{C}\to 0_{+}$, this means that *A* and *B* are infinitely close to the same point *C* from both sides. Then, the slope of point *C* on the $v_{p}^{2}-S$ curve is 2a. To meet the constraints of the ground vehicle, the condition $2a_{x-}\leq 2a\leq 2a_{x+}$ must be guaranteed.

Algorithm 2 presents the overall workflow of the route-plane trajectory planning. The basic idea is to maximize the velocity without violating the velocity and acceleration limits at each point, which will accordingly minimize the total travel time. First, the slope suppression function LinePress is defined to ensure that the squared velocity is suppressed below a set value from the starting point to satisfy the upper and lower limits on the acceleration. Then, the starting point, ending point, point where the acceleration exceeds the upper limit, and point where the acceleration exceeds the lower limit are processed separately so that the linear velocity is maximized while neither the velocity nor the acceleration violates their limits at each point.

After the squared velocity $v^{2}$ is calculated for each point on the road line, the time stamp *t* of each point is calculated according to $t\left(s\right)=\int_{0}^{s}\frac{1}{v(S)}dS$. The lateral acceleration $a_{s}$ of each point is calculated according to $a_{s}=\frac{dv}{dt}=\frac{1}{2}\cdot \frac{d(v^{2})}{dS}$. The longitudinal acceleration $a_{n}$ is calculated according to $a_{n}=\frac{v^{2}}{r}$. The trajectory planning is then complete.    
**Algorithm 2:** Route-plane Trajectory Planning**Input**: $v_{p}^{2}\left(u\right)$, S(u), $a_{x+}$, $a_{x-}$**Output**: $v^{2}\left(u\right)$‒‒‒‒‒‒‒‒‒‒‒‒‒‒‒‒‒‒‒‒‒‒‒‒‒‒‒‒‒‒‒‒‒‒‒‒‒‒‒‒‒‒‒‒‒‒‒‒‒‒‒‒‒‒‒‒‒‒‒‒‒‒‒‒‒‒‒‒‒‒‒‒‒‒‒‒‒‒‒‒‒‒‒‒;**Function Y= LinePress**($Y,X,y,X_{i},K$);**if** K>0**then**
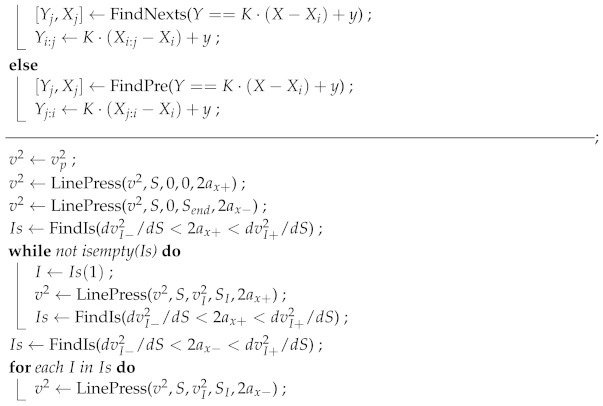
return $v^{2}$

## 8. Experiments

For our practical experiments, we carefully chose a specific location in Jinnan District, Tianjin, China, which spanned an area measuring 80 m in length and 60 m in width in Figure 10a. To extract precise data pertaining to the road surface, we leveraged the pixel information derived from aerial images. These data were subsequently processed to obtain discrete values, which were then fitted to form a road line comprising segmented Bézier curves. The trajectory planning was meticulously executed, taking into consideration the operational capabilities of a real-world ground vehicle. This approach ensured a comprehensive and accurate representation of the road environment during our experimental endeavors.

In this particular instance, we establish the initial point of the road as the origin of our data. We then proceed with a translation operation, followed by a rotation process based on the disparity between the image coordinate system and the world coordinate system. Additionally, we scale down the image coordinate data and external references to ensure compatibility. Figure 10b illustrates the original data as black discrete points, while the red discrete points represent the corrected data. Moving forward, in Figure 10c, we observe the successful transfer of data from the image coordinate system to the world coordinate system. At this stage, the road information contained within the image is effectively extracted and transformed into discrete points representing the road surface within the world coordinate system. This meticulous process ensures accurate alignment and facilitates meaningful analysis of the road information within the broader context of the world coordinate system.

Figure 10d presents the outcomes of road simulation utilizing the conventional road geometric parameter model. This model, widely employed in road engineering, encompasses various geometric parameters that define the road’s shape and layout. As shown in Figure 10e, we measured several road boundary points in the global coordinate system and plotted them with the star points and road model. It can be seen from Table 2 that the piecewise Bézier road fitting model based on the genetic algorithm proposed in this paper is superior to the traditional method in terms of fitting effect and running speed.

To achieve enhanced precision in parameter fitting, we employed a method wherein we captured streaming video footage while the aerial vehicle remained stationary, thereby obtaining multiple frames of an identical scene. Subsequently, we subjected these multiple video frames to iterative processing using the road extraction and fitting algorithm. Through this iterative approach, we were able to evaluate the performance metrics of each frame and select the one that exhibited the most favorable outcomes as the final result. This meticulous process ensured that the chosen frame accurately represented the road environment, thereby contributing to the overall reliability and validity of our experimental findings. Figure 10f shows the results for a frame with the best fitting performance index.

To comply with the performance requirements of unmanned ground vehicles, we selected the following constraints on the velocity and acceleration: $v_{M}=2$ m/s, $v_{m}=1.5$ m/s, $a_{x+}=1$ m/s${}^{2}$, $a_{x-}=-0.8$ m/s${}^{2}$, $f_{y_{\max}}/M=0.02$ m/s${}^{2}$.

We used Algorithm 2 for planning, and the resulting trajectory is shown in Figure 11. Figure 11a shows the route-plane trajectory, and Figure 11b shows the resulting curve in the t−S−V space. The v−t, v−S, and S−t curves were obtained for the observation planes along the three coordinate axes. Figure 11c shows the curve in the $t-S-a_{s}$ space. The $a_{s}-t$, $a_{s}-S$, and S−t curves were obtained along the three coordinate axes. Figure 11d shows the S−t curve. The total path length was 104.3 m, the travel time *T*—travel was 125.1 s, and the average velocity was 0.834 m/s.

## 9. Conclusions

We propose a sorting method based on the K-Means clustering algorithm and a piecewise road vectorization based on the GA-Bézier algorithm to extract and fit roads from aerial images. The road model can then be used with a dynamic model of the ground vehicle for route-plane trajectory planning to minimize the travel time by maximizing the velocity of the ground vehicle during turns. Practical experiments were performed to verify the effectiveness of the method. Future work will involve incorporating the height information into the road extraction and using 3D reconstruction technology and machine learning technology [22] to obtain the road in 3D space and match the actual scene more closely. At the same time, real-time detection of drones is also a very important link. IoT robots with artificial intelligence can be applied to various surveillance fields [23]. In the design and implementation direction of real-time obstacle detection and obstacle-avoiding mobile robots [24], the Internet of Things and machine learning can give us great help.

## Figures and Tables

**Figure 1 sensors-23-07305-f001:**
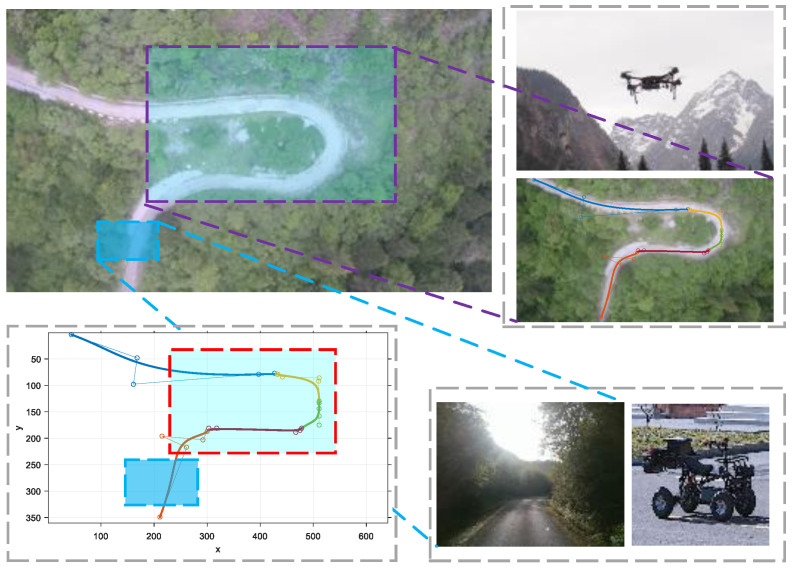
Using aerial vehicles to visually guide the trajectory of unmanned ground vehicles on unknown curved roads. The aerial images are used to extract and model roads, which are then used to plan trajectories that satisfy the constraints of a continuous tangent and continuous curvature.

**Figure 2 sensors-23-07305-f002:**
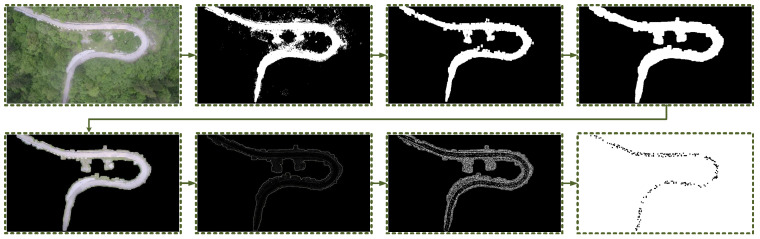
Method of extracting star points from an image of a road: brightness thresholding, morphological opening operation filtering, obtaining the ROI contour, extracting the ROI area, Laplacian texture sharpening, sharpening image binarization, and obtaining the star points.

**Figure 3 sensors-23-07305-f003:**
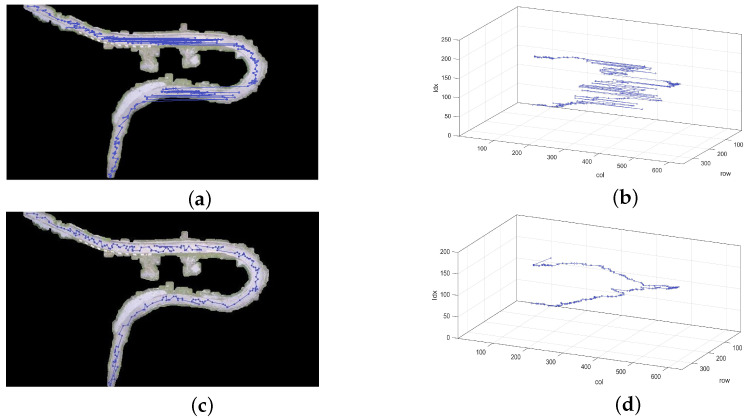
Star point order. (**a**,**b**) The top images show the default order. (**c**,**d**) The bottom images show the sorted order.

**Figure 4 sensors-23-07305-f004:**
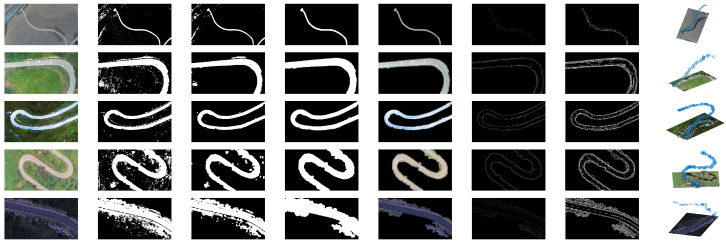
Road extraction results. The sub-images in each column show the original image, brightness thresholding result, morphological processing result, ROI contour, the ROI area, texture sharpening map, sharpening image binarization result, and road star point sorting result.

**Figure 5 sensors-23-07305-f005:**
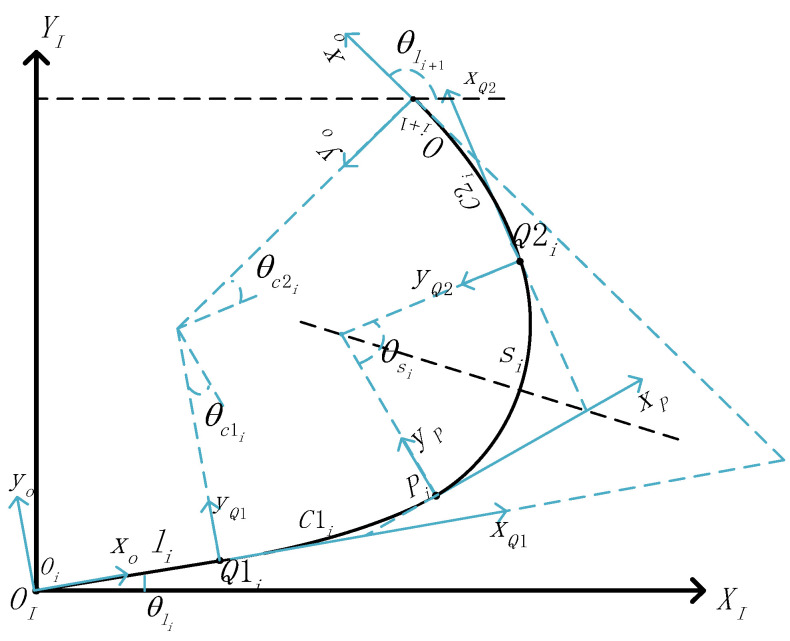
Route unit model.

**Figure 6 sensors-23-07305-f006:**
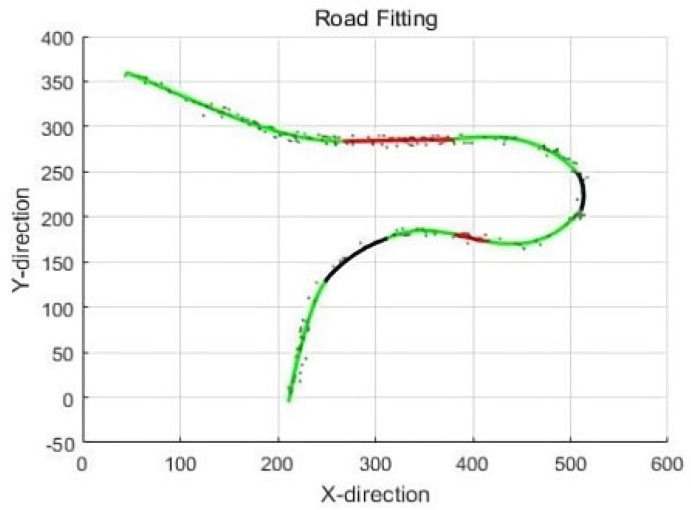
Sichuan road route extraction effect map.

**Figure 7 sensors-23-07305-f007:**
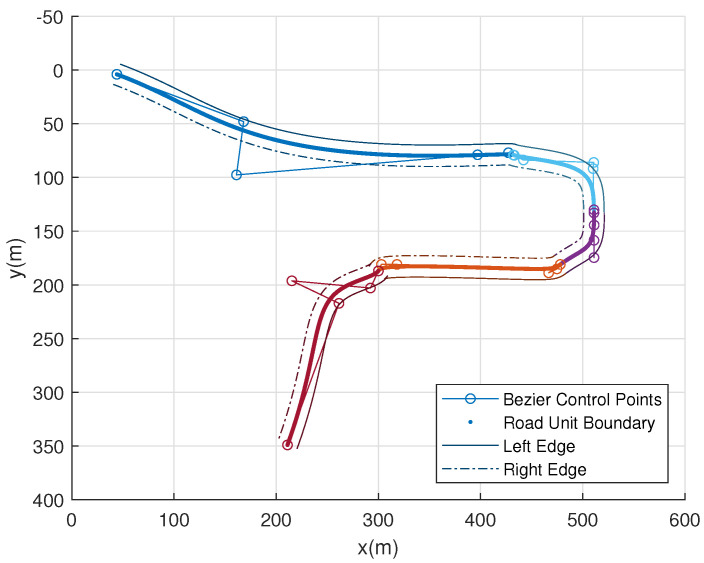
Piecewise road vectorization model. Road units are added according to the constraints of a continuous tangent and continuous curvature at junctions to obtain the overall road.

**Figure 8 sensors-23-07305-f008:**
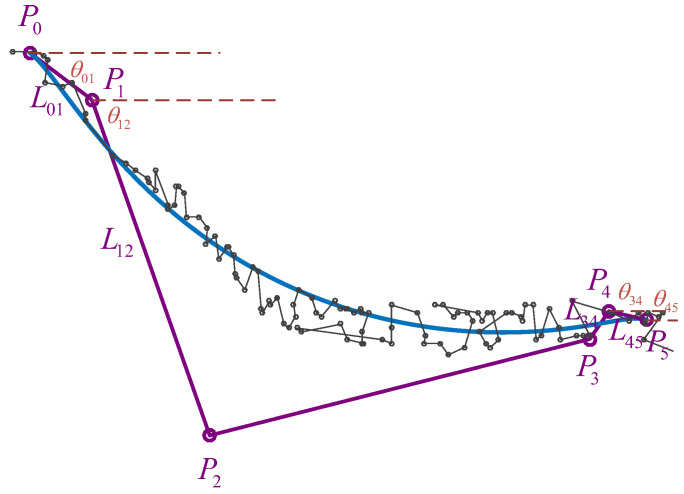
Fitting scheme for the first road unit. Because only the positions of the starting and ending points are constrained, the chord length and angle between adjacent control points can be used as optimization variables to solve. Then, the coordinates of the control points can be calculated.

**Figure 9 sensors-23-07305-f009:**
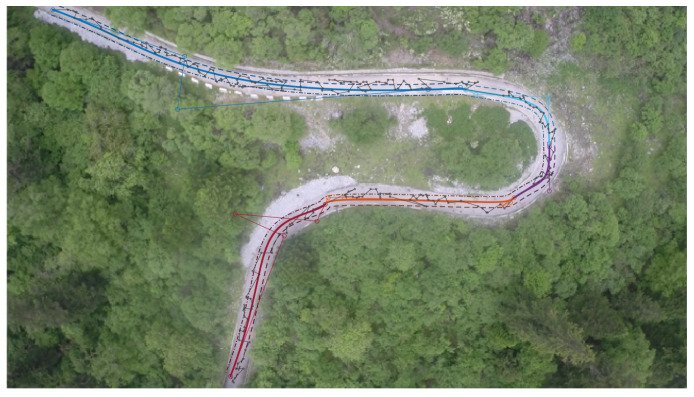
Road line fitting results. The thick solid line represents the centerline of the road, and different colors are used to distinguish different road units. The large colored circles represent the control points of each road unit.

**Figure 10 sensors-23-07305-f010:**
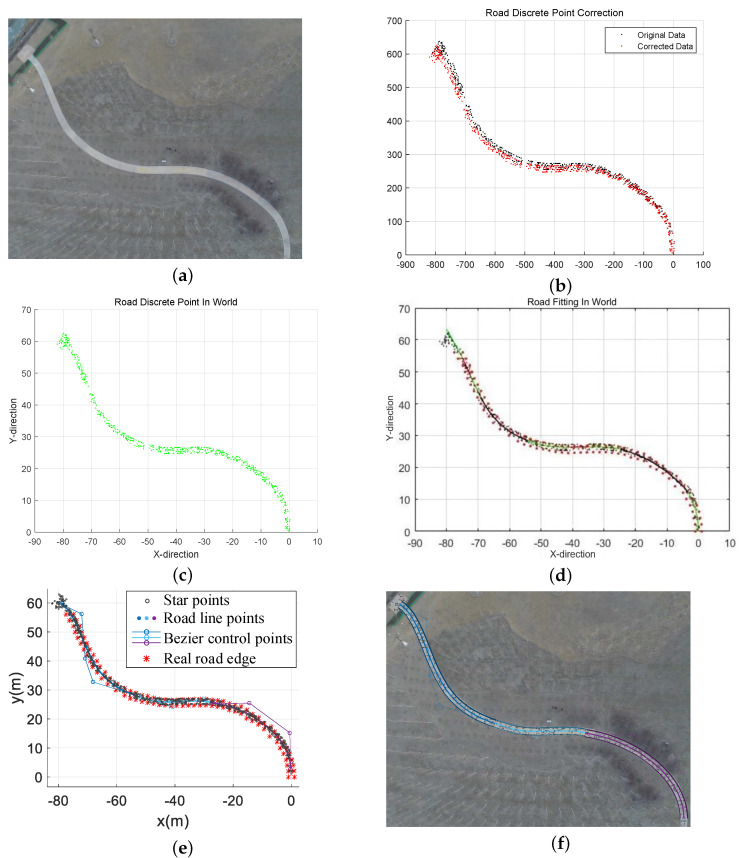
(**a**) Nankai road. (**b**) Discrete point data correction. (**c**) Discrete points in the world coordinate system. (**d**) Route fitting based on the traditional method. (**e**) Route fitting based on the proposed method. (**f**) Route extraction effect map.

**Figure 11 sensors-23-07305-f011:**
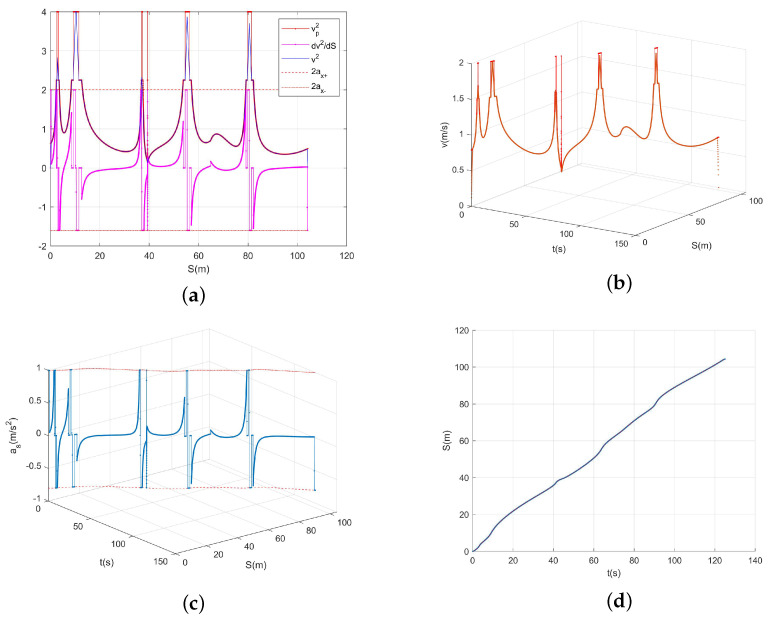
(**a**) Results of the route-plane trajectory planning. (**b**) The planning results in the velocity space. (**c**) The planning results in the acceleration space. (**d**) The last image shows which waypoints should be reached at each timestamp.

**Table 1 sensors-23-07305-t001:** Road geometry parameters.

*i*	*L*/(m)	$\theta_{c1}$/(°)	*R*/(°)	$\theta_{s}$/(°)	$\theta_{c2}$/(°)	±1
1	0	17.88	143.43	32.19	42.22	−1.00
2	33.85	64.95	54.02	45.06	52.97	1.00
3	112.77	66.74	0	0	0	−1.00

**Table 2 sensors-23-07305-t002:** Comparison of road fitting performance based on two methods.

Road Fitting Method	EV/(m)	*T*—Model/(s)
Traditional	21.36	1.37
Proposed	19.04	0.13

## Data Availability

Not applicable.

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
