# Peer review of "A Novel Vectorized Curved Road Representation Based Aerial Guided Unmanned Vehicle Trajectory Planning"

_sensors, 2023, doi:10.3390/s23167305_

Round 1

Reviewer 1 Report

This is an interesting paper with a beautiful approach to solve a problem. I have some remarks and suggestions to improve the paper, of which the first two appear crucial to me

The problem/use case is not clear to me. In which concrete situations is there a need for automated driving without GPS? This needs to be repeated/stressed since currently  the paper describes an interesting technical/mathematical solution, however the use cases where this is necessary/desirable and how this could be implemented in practice are not discussed. In my opinion, the work does not make sense without a correct understanding of this fact and I recommend elaborating on this.

I’m confused about the following elements of the study: Figure 2: The image processing steps are illustrated. Are these calculations to be done on the UAV? Is this computational capacity available (and not problematic for the UAV’s autonomy on battery)? P2l35-37 the authors indicate that a wireless transmission of images is a problem, however I don’t understand why this is the case (even commercial UAVs have excellent image transmission capabilities). In the experiment, p12l292, I also get the impression that images were streamed from the drone to a ground station that did all of the calculations? This contradicts contribution 1 p2 l45-47 where it is stated that this solution will result in a reduction of information that needs to be transmitted. Please clarify all of this in the text.

Contribution 1 p2 l45-47: the results of the experiments don’t support this claim: the authors should make a comparison in order to claim this.

It is assumed that a geo-calibrated reference points are available. This makes me wonder if this effort needs to be spent, aren’t there other ways of navigation with an equal effort?

In remote areas where this application could be useful, I can imagine that the road surface quality may not be optimal at all locations. Therefore, I feel that it is necessary to include some kind of safety margin in the path planning phase to make sure the vehicle’s maximum speed and acceleration does not exceed safe levels when cornering.

Perhaps it is more intuitive to speak of ‘lateral’ and ‘longitudinal’ (tangential) acceleration instead of ‘radial’ and ‘normal’ (which are mathematical equivalents).

How does it perform when there are occlusions from vegetation (overhanging trees etc.)?

Proof-reading by a native English speaker is recommended to improve readability.

Author Response

We appreciate the time and effort that you have dedicated to providing your valuable feedbacks on our manuscript. Based on these comments, we have made careful revisions which we hope could meet with your approvals. Additionally, in this response, all the text quoted from the original submitted paper is marked in “blue”, while all the revised parts are in “orange”. Any revisions to the manuscript have been highlighted in red.

Reviewer 2 Report

The detailed content is presented in the supplementary word document.

Author Response

(The authors gave the same response as above.)

Reviewer 3 Report

Dear Authors,

I have reviewed the manuscript “Piecewise GA-Bézier Algorithm Based Vectorized Curved Road Representation for Aerial Guided Unmanned Vehicle Trajectory Planning” Manuscript ID: sensors-2495910 that has been submitted for publication in the: Sensors (ISSN 1424-8220) and I have identified a series of aspects that in my opinion must be addressed to bring a benefit to the manuscript.

The article under review will be improved if the authors address the following aspects in the text of the manuscript:

1.     The abstract does not provide enough technical details about the novel air-ground collaborative vectorized curved road representation and trajectory planning method. It would be beneficial to include information about the specific algorithms or techniques employed, as well as any unique features or advantages of the proposed method.

2.     While the abstract mentions that simulations and experiments were conducted to verify the effectiveness of the proposed method, it does not provide any specific results or findings from these evaluations. Including key performance metrics or outcomes would enhance the understanding of the method's effectiveness.

3.     It is preferable to use more than one dataset to confirm the effectiveness of the proposed method used.

4.     The paper does not include a thorough comparative analysis of the proposed algorithm with existing methods or state-of-the-art approaches. Comparative evaluations are essential to demonstrate the advantages and improvements offered by the proposed algorithm over other techniques. Without such analysis, it becomes challenging for readers to assess the novelty and effectiveness of the proposed method.

5.     The authors do not address the limitations or potential challenges of the proposed algorithm. Identifying and discussing the limitations, constraints, or scenarios in which the algorithm may not perform optimally would provide a more balanced view of its applicability and help readers understand its potential shortcomings.

6.     The references need to be updated for the years 2022 and 2023, as this field has been recently raised.

https://doi.org/10.1016/j.cscm.2022.e01463

Author Response

(The authors gave the same response as above.)

Reviewer 4 Report

Authors presented their idea through "Piecewise GA-Bézier Algorithm Based Vectorized Curved Road Representation for Aerial Guided Unmanned Vehicle Trajectory Planning" which seems a long long title. And it looks like authors have mentioned each and everything in title only.

The proposal presented seems less of novelty as the roads are made up with same concepts in hills especially, then what is the inventiveness in this ?

If you are using a predesigned mathematical model then what is your contribution here?

Some recent relevant papers must be seen by authors like a) Design and implementation of an ML and IoT based adaptive traffic-management system for smart cities b) Internet of Things: Robotic and Drone Technology

The equations are not defined/explained well in the paper. Further, the comparison with recent work to prove the effectiveness of proposal must be there.

Abstract seems without any storyline for signifying the need of this work.

At some places the grammar use of English is not considered.

Author Response

(The authors gave the same response as above.)

Round 2

Reviewer 2 Report

1.  The revised paper presents the potential approaches for the realization of implementation more clearly, enhancing the persuasiveness of the feasibility of this method.
2.  A more comprehensive literature review and method comparison have been conducted, elevating the cutting-edge nature of this article.
3.  The introduction of clustering algorithms prior to sorting has led to an improvement in both the fitting quality and running time of road fitting method.
4.  Figure 10 (d) appears to have a different resolution compared to other legends in Figure 10. If possible, it would be preferable to replace it with a higher-resolution image.

Author Response

Thanks for your helpful comments. We will keep persistent efforts in further exploration of road modeling and vehicle trajectory planning problems.

Reviewer 3 Report

Accept in present form

Author Response

(The authors gave the same response as above.)

Reviewer 4 Report

Paper seems acceptable.

Author Response

(The authors gave the same response as above.)
